# Preparation of long single-strand DNA concatemers for high-level fluorescence in situ hybridization

Dongjian Cao[1,3], Sa Wu[1,3], Caili Xi[1], Dong Li[1], Kaiheng Zhu[1], Zhihong Zhang[1], Hui Gong[1], Qingming Luo [1,2,4✉] &
Jie Yang [1,4✉]

Fluorescence in situ hybridization (FISH) is a powerful tool to visualize transcripts in fixed cells and tissues. Despite the recent advances in FISH detection methods, it remains challenging to achieve high-level FISH imaging with a simple workflow. Here, we introduce a method to prepare long single-strand DNA concatemers (lssDNAc) through a controllable rolling-circle amplification (CRCA). Prepared lssDNAcs are used to develop AmpFISH workflows. In addition, we present its applications in different scenarios. AmpFISH shows the following advantages: 1) enhanced FISH signal-to-noise ratio (SNR) up to 160-fold compared with single-molecule FISH; 2) simultaneous detection of FISH signals and fluorescent proteins or immunofluorescence (IF) in tissues; 3) simple workflows; and 4) cost-efficiency. In brief, AmpFISH provides convenient and versatile tools for sensitive RNA/DNA detection and to gain useful information on cellular molecules using simple workflows.

[1] Britton Chance Center and MoE Key Laboratory for Biomedical Photonics, School of Engineering Sciences, Wuhan National Laboratory for Optoelectronics-Huazhong University of Science and Technology, Wuhan, Hubei 430074, China. [2] School of Biomedical Engineering, Hainan University, Haikou 570228, China. [3]These authors contributed equally: Dongjian Cao, Sa Wu. [4]These authors jointly supervised this work: Qingming Luo, Jie Yang. ✉email: qluo@mail.hust.edu.cn; yangjie@mail.hust.edu.cn

Complex biological tissues consist of different types of cells. Characterizing the temporal and spatial heterogeneities of RNA or DNA is critically important to understand cellular functions. Fluorescence in situ hybridization (FISH), a powerful tool, allows the detection of nucleic acid molecules within the intact structure and in the morphology of cells or tissue sections. FISH has been widely applied in the research[1–3] and clinical diagnostic assays[4,5]. The readout of the FISH signal can be achieved using fluorophore-tagged oligonucleotide probes that bind with specific nucleic acid sequences by complementary pairing. Therefore, the spatial and quantitative information of nucleic acids can be detected in the cells or tissues under a fluorescent microscope using single-molecule RNA fluorescence in situ hybridization (smFISH) imaging, which allows visualizing the spatial distribution of transcripts through hybridization of target genes[6]. Because it is difficult to detect individual fluorophores by conventional fluorescence microscopy, smFISH usually requires multiple oligonucleotide probes with fluorophore to target different regions of RNA or DNA[7] to acquire detectable signals. Despite the accumulation of multiple fluorophore molecules to a certain extent, the accuracy of the smFISH assay is still affected by the low signal intensity with high background signals from the tissue. Moreover, smFISH protocols usually are limited to long RNA molecules (>3 kbp) that can provide more fluorescent-binding sites to achieve enough detectable signals[8].

The literature reports several amplification strategies for FISH, such as hybrid chain reaction (HCR)[9,10], branch DNA (bDNA)[11,12], ClampFISH[13], and SABER[14]. Among these, HCR allows in situ amplification via folding hairpin oligonucleotide pairs that trigger a self-assembly chain. However, HCR may not offer sufficient sensitivity in certain cases[15], and designing hairpin structures is a time-consuming process. Moreover, there is a lack of systematic guidelines for hairpin design[16]. All other amplification strategies for FISH can provide high signal levels; however, they require the synthesis of more expensive modified oligonucleotides, such as bDNA[11,12], azide–alkyne[13], and inverted bases[14]. An ideal FISH imaging should display both cost-efficiency and bright signals that can potentially shorten the exposure time for imaging, thereby achieving high accuracy at low costs.

Here, we established a more cost-efficient strategy for high-level FISH signals. First, a method was prepared to develop a long single-strand DNA concatemer (lssDNAc) through controllable rolling-circle amplification (CRCA). Second, based on lssDNAc, several easily implemented workflows of AmpFISH were established. Lastly, we used simple AmpFISH workflows to specifically detect transcripts or chromosomes in the cells and tissues. Our results showed that AmpFISH has several advantages: (1) up to 100-fold signal-to-noise amplification; (2) simultaneous detection of FISH signals with fluorescent proteins or immunofluorescence (IF) in tissues; (3) simple workflow; and (4) cost-efficiency. To sum up, we present a convenient and versatile tool for FISH detection, providing a promising application for transcript and chromosome assays in the cells or in complex and heterogeneous tissues.

## Results

### Design of amplification method and preparation of hybrid probes for AmpFISH.
To establish a pipeline for FISH assay, we developed a preparation method for lssDNAc using rolling-circle amplification in vitro. Here, the concatemer sequence provides multiple repeat sites for binding multiple fluorophore-tagged oligonucleotide probes. The accumulated probes can effectively amplify the FISH signal.

First, we designed two partially complementary DNA oligonucleotides, named padlock and adaptor, as shown in Fig. 1a. At the 5′ terminus of the adaptor, we designed a toehold sequence that was complementary to target molecules. The sequences of 14–16 nucleotides at the right arm (B1) and ten nucleotides at the left arm (B2) of the padlock were complementary to the adaptor, respectively. After annealing to the adaptor, the right arm (5′ phosphorylation terminus) and the left arm (3′-OH terminus) of the padlock formed adjacent nicking sites when they were complementary with the 3′ terminus of the adaptor. Therefore, the right and left arms of the padlock could be ligated by DNA ligase to form a circular ssDNA. When DNA polymerase and dNTP were added to the reaction system, the ssDNA of the adaptor could be extended by displacement amplification, as shown in Fig. 1a.

To establish a user-friendly reaction, we used an all-in-one tube strategy to amplify adaptor ssDNA (see "Methods"). After padlock and adaptor were annealed, T4 DNA ligase was used to ligate the nicking site of the padlock. To extend the adaptor ssDNA, we utilized DNA polymerases with displacement ability using the rolling circle. At first, two kinds of DNA displacement polymerases, phi29 and Bst 2.0, were tested. After extended by DNA polymerases, the products were run through agarose gel electrophoresis, as shown in Fig. 1b. Both DNA polymerases could extend to form clear amplification bands in the gel. Phi29 DNA polymerase provided rapid extension, whose amplification length reached up to 10 kb (amount to dsDNA marker) in only 5 min, and its extension speed was consistent with that reported in the previous literature[17]. However, overlong DNA molecules extended by phi29 DNA polymerase were not suitable for FISH assay, because a molecule with a large molecular weight results in slow diffusion and poor penetration ability, and could affect the efficiency of FISH[18]. Therefore, we excluded phi29 DNA polymerase in subsequent experiments. In contrast, Bst 2.0 DNA polymerase showed relatively slower amplification at 50 °C, which extended approximately 0.5 kb in length after 1 h (Fig. 1b). To investigate its extension ability under different conditions, we amplified the ssDNA adaptor under three reaction temperatures (45, 50, and 55 °C) with three reaction times (0.5, 1, and 2 h). The results of the electrophoresis assay showed that their extension length increased slightly as the time or temperatures increased, indicating that Bst 2.0 DNA polymerase provided a relatively controllable extension (Fig. 1c). Therefore, the lssDNAc lengths can be more easily modulated by changing only the reaction time. Similar to Bst 2.0 DNA polymerases, Klenow (exo⁻) exhibited the ability of controlled extension at 37 °C (Supplementary Fig. 1). After gel electrophoresis, we recycled the extended lssDNAcs from agarose gel using a gel extraction kit to obtain purified extended ssDNAc for the FISH assay. The schematic representation of lssDNAc preparation workflows is depicted in Fig. 1d.

### AmpFISH effectively amplifies fluorescent signals in cultured cells.
To evaluate the feasibility of lssDNAc as a FISH probe, we first used AmpFISH in HeLa cells that were transiently transfected with the pCAG-EYFP plasmid. A set of probes containing four lssDNAcs with a length of ~750 bp could target EYFP mRNA. They were mixed and added to the cells transiently transfected with the pCAG-EYFP plasmid. A workflow of AmpFISH that could retain fluorescent protein for culture cells was developed, allowing simultaneous detection of EYFP protein and AmpFISH for *EYFP* mRNA (see "Methods" and Supplementary Fig. 2, left panel). The imaging results showed that the AmpFISH signal perfectly localized to the cells positive for EYFP fluorescence, whereas cells with negative EYFP fluorescence did not exhibit FISH signal (Fig. 2a). These results indicate that lssDNAc can be effectively used to assess EYFP mRNA expression with high specificity in cultured cells. As shown in Fig. 2a, intensities of the AmpFISH signal were not completely consistent with those of EYFP. We presume that the turnover pattern of

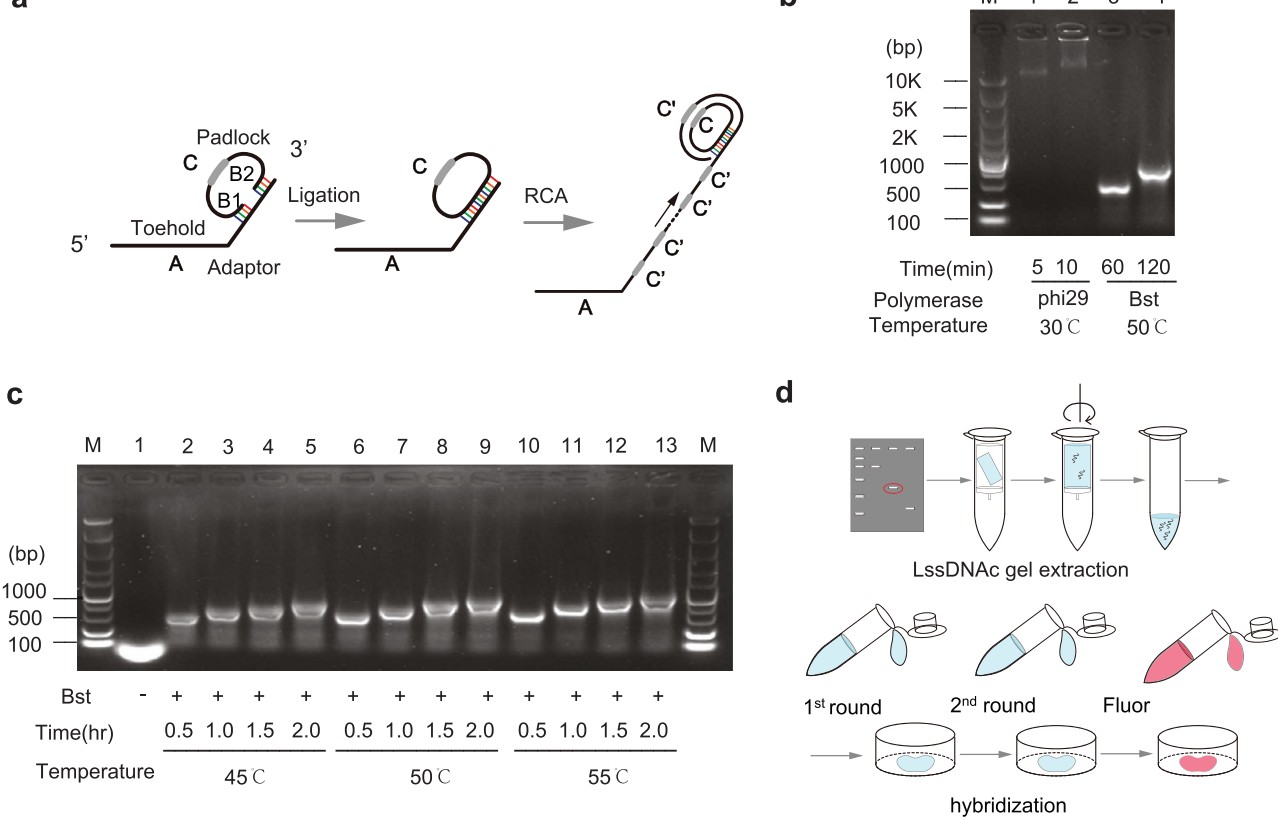

**Fig. 1 Preparation of lssDNAc. a** Schematic representation of the extension reaction for lssDNAc. Both nucleotides of 5′ and 3′ termini of the padlock can complementarily bind to the 3′ termini of the adaptor. Next, the padlock is ligated at 5′ and 3′ termini by DNA ligase to form a closed circle. The adaptor ssDNA can be extended using the closed padlock as a template using RCA. **b** The amplification products were assayed via agarose gel electrophoresis. The extended products of adaptor ssDNA were amplified by phi29 DNA polymerase (lane 1 for 5 min, lane 2 for 10 min) and Bst 2.0 DNA polymerase (lane 3 for 1 h, lane 4 for 2 h). **c** The ssDNA adaptors were amplified to form products of different lengths by Bst 2.0 DNA polymerase (lanes 2–14) under three reaction temperatures (45 °C for lanes 2–5, 50 °C for lanes 6–9, and 55 °C for lanes 10–13) with different amplification times (0.5, 1, 1.5, and 2 h), and in the absence of DNA polymerase (lane 1). **d** The schematic diagram depicting the preparation for lssDNAc and hybridization workflow.

rapid mRNA[19] was different from that of EYFP protein. To further evaluate the sensitivity and specificity of AmpFISH, we selected four transcripts (*Actinβ*, *Lox*, *Txn1*, and *Sla2*) with different expression levels in NIH3T3 cell line from the UMI-RNAseq datasets, one of the absolute quantitative transcriptome sequencing technology[20] (see the "Methods" section for details). We conducted AmpFISH assay for those four transcripts in NIH3T3 cell line (Fig. 2b), whose results showed a consistent tendency for the four transcripts' abundance compared to those measured using the UMI-RNAseq method (Fig. 2c). Although the measure abundance of *Actinβ* via AmpFISH assaying is relatively lesser than that via UMI-RNAseq, we speculate overcrowded AmpFISH puncta of *Actinβ* should result in less counting.

To quantitatively compare the intensities or signal-to-noise ratio (SNR) of AmpFISH, we performed a series of FISH experiments for exogenous *Cas9* mRNA assay in stable cell lines transfected with the pX330-Cas9–2A-Cerulean-2A-PuroR plasmid (Supplementary Fig. 3). Conventional smFISH (without extension), primary, secondary, and tertiary amplifications were performed (Fig. 2d). Among these, the secondary amplification included branched structures that were formed by secondary concatemers by binding to primary concatemers. Tertiary amplification formed high-level branching based on secondary amplification, as shown in the upper panel of Fig. 2d. We quantified and compared the signals of different amplifications. The results showed that smFISH presented an extremely faint signal (Fig. 2d, the rightmost panel). Compared with the SNR

signal of smFISH, the primary, secondary, and tertiary amplification AmpFISH were enhanced by 18.9-fold, 33.2-fold, and 168.3-fold improvement, respectively (Fig. 2d, e).

**AmpFISH enables robust chromosome assay**. To check the feasibility of AmpFISH in chromosome assays, we detected the chromosome ploidy of culture cells (Supplementary Fig. 2, right panel). Aneuploidy alterations are hallmarks of tumor aggressiveness and increase with malignant progression[21]. The chromosome ploidy can be used for clinical cancer assay. Because centromere satellite repeats provide multiple binding sites, we only used one lssDNAc to target the centromere of chromosome 8. After hybridization, tomography of the cell nucleus was performed by confocal microscopy (Supplementary Movie 1), and subsequently, three-dimensional (3D) images were merged, as shown in Fig. 3. The results showed that most of the HeLa cells have a triploid karyotype of chromosome 8, which is consistent with that reported in a previous study[22]. The results revealed that AmpFISH efficiently labeled interphase chromosomes of HeLa cells and is a promising assay for assessing chromosome ploidy.

**AmpFISH enables robust transcript assay in tissues**. Next, we assessed whether AmpFISH could robustly assay transcripts in tissues, apart from cultured cells. We used mouse brain tissue to check the feasibility of AmpFISH for tissue assay. At first, we tested the primary amplification probes for *Gad1* mRNA via methanol penetration. However, the primary amplification

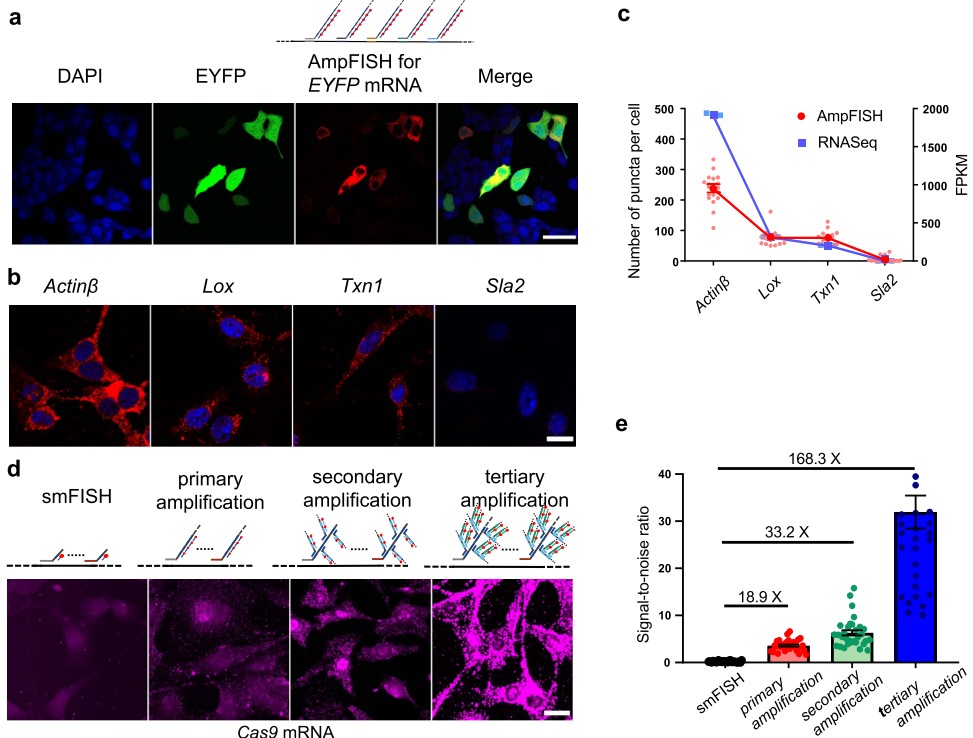

**Fig. 2 AmpFISH facilitates labeling of transcripts with high specificity and sensitivity (a–c) and the effective amplification of fluorescent signals (d–e) in cultured cells. a** A set of primary amplification probes for targeting the EYFP mRNA was used in HeLa cells transiently expressing EYFP. FISH signal (red, signal labeled by Cy5, the third panel from left in (**a**) and fluorescent protein signal (green signal in EYFP panel, the second panel in (**a**) were imaged in the same field of view (scale bar: 50 μm). **b** The four endogenous mRNAs (*Actin*β, *Lox*, *Txn1*, and *Sla2*) were assayed using AmpFISH imaging in NIH3T3 cell lines (scale bar: 20 μm). **c** The quantitative analysis signal puncta in (**b**) of the four endogenous mRNAs in NIH3T3 cells of AmpFISH puncta per cell are shown. The mean ± SEM of AmpFISH puncta per cell are shown using red round dots, $n_{(Actinβ)} = 20$; $n_{(Lox)} = 15$; $n_{(Txn1)} = 15$, and $n_{(Sla2)} = 19$. The four transcripts from the RNAseq dataset (blue line and square dots) were compared. Each dot represents the mean ± SEM ($n = 3$); FPKM denotes Fragments Per Kilobase Million. These results illustrate the similar relative transcript abundance between the different methods. **d** Fluorescent signal labeled by TAMRA dye was amplified by AmpFISH in stable NIH3T3/cas9 cells. The confocal imaging for smFISH (unextended probe), primary, secondary, and tertiary amplifications are arranged from left to right, respectively (scale bar: 50 μm). **e** Comparative analysis of the signal-to-noise ratio of different amplification hierarchies. The intensity of the background signal, from an area adjacent to the area from which the signal was recorded, was subtracted from the intensity of the signal. The signal-to-noise ratio was determined using the formula: (Signal-background)/background.

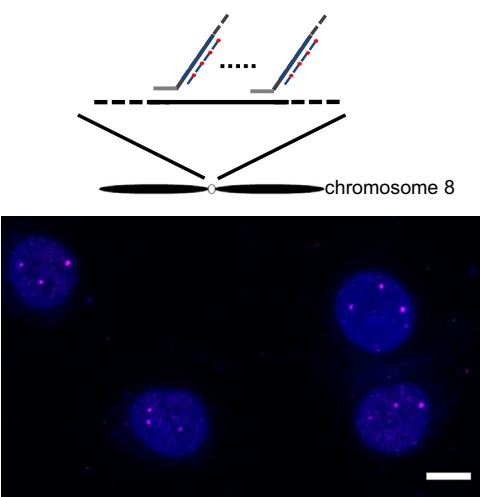

**Fig. 3 AmpFISH efficiently labeled interphase chromosomes of HeLa cells using primary amplification. Most centromeres of HeLa chromosome 8 show three FISH signal puncta.** Scale bar: 25 μm.

probes provided an extremely faint signal with a high tissue background (Supplementary Fig. 4a). Next, further branching amplification via secondary amplification probes was performed. The secondary amplification enhanced the FISH signal compared with primary amplification (Supplementary Fig. 4b).

Given that brain tissue is rich in lipid components that could contribute to a high-signal background[23], we optimized AmpFISH workflows to clear the background. We evaluated varying concentrations of sodium dodecyl sulfate (SDS) and methanol permeabilization (with/without) in four experimental workflows, as shown in Fig. 4a. The signal intensity and the SNR from four workflows were compared (Fig. 4b, c). Treatment with 0.2% or 1% SDS and methanol permeabilization improved SNR by three to four folds when compared with no SDS and methanol treatment and by ~two folds when compared with methanol treatment only. For further optimization, we adjusted the paraformaldehyde (PFA) post-fixation time (Supplementary Fig. 5) and the hybridization time of primary amplification probes (Supplementary Fig. 6) in the mouse brain slices. We found that a high SNR in the mouse brain slice was produced without post-fixation and primary amplification hybridization for 24 h.

To further verify the accuracy of our experimental workflow, we detected EGFP mRNA using methanol and 1% SDS treatment described above in brain tissue slices of *Thy1*-EGFP transgenic

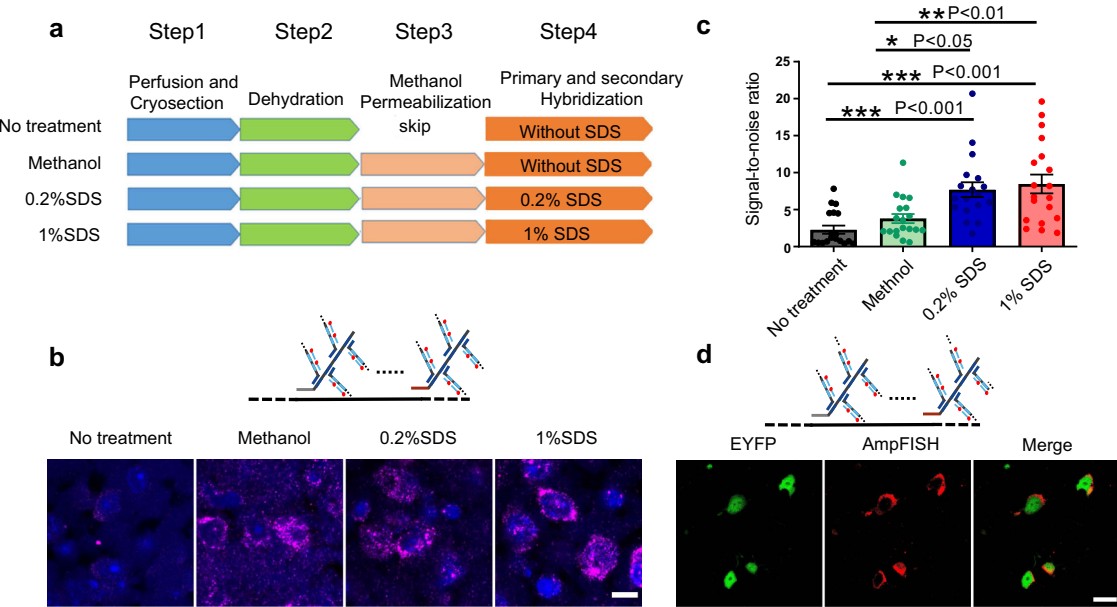

**Fig. 4 AmpFISH is adopted to robustly assay transcripts in mouse brain slices. a** Schematic diagram depicting four different experimental workflows. **b** FISH imaging of *Gad1* gene in mouse brain slices using different pipelines depicted in (**a**) (scale bar: 10 μm). **c** Statistical analysis of signal-to-noise ratio (SNR) of AmpFISH from four experimental pipelines in (**b**). The intensity of an area as background signal next to the signal was subtracted from the signal itself. The signal-to-noise ratio was determined by formula: (Signal-background)/background. One-way analysis of variance (ANOVA), $F(3, 72) = 10.83$, $P < 0.0001$; Bonferroni's multiple comparison test, no treatment vs. 0.2% SDS, $t = 4.206$, $P < 0.001$; no treatment vs. 1% SDS, $t = 4.810$, $P < 0.001$; methanol vs. 0.2% SDS, $t = 3.030$, $P < 0.05$; methanol vs. 1% SDS, $t = 3.633$, $P < 0.01$. ($n = 19$, from three repeat experiments). **d** The specificity of AmpFISH via secondary amplification hybridization was verified in brain slices by expressing EYFP protein in *Thy*-EGFP transgenic mouse (scale bar: 25 μm).

mouse. Figure 4d (left panel) shows that the fluorescence signal of EGFP was retained through treatment with 1% SDS and 100% methanol, allowing to assess the accuracy of AmpFISH by analyzing the distribution of AmpFISH signal and fluorescence signal from proteins in the same slice. Figure 4d (middle and right panels) show that the AmpFISH signal (red) perfectly matched with the EGFP fluorescence (green) in the same cells, demonstrating that the AmpFISH workflow that we developed exhibited specificity in the brain slices.

**AmpFISH in combination with IF in brain slices**. Immunofluorescence (IF) can visualize cellular antigens such as proteins based on the antigen–antibody reaction. IF is extensively used in the field of clinical diagnostics, research, and drug development. To study multiple applications for AmpFISH, we optimized the AmpFISH workflow to make it compatible with IF. Briefly, SDS was replaced by HCl (0.1 N), proteinase K (1.5 μg/mL), and 2% $H_2O_2$ in the optimized workflow to treat the tissues (see "Methods"). To assess the sensitivity of AmpFISH and IF under the optimized workflow, we simultaneously detected protein and mRNA of calcium/calmodulin-dependent protein kinase IIα (CaMKIIα) by a combination of IF and AmpFISH in the same slices from the caudate putamen (CPu) area of the brain. As shown in the upper panels of Fig. 5a, up to 90% of AmpFISH (red) and IF signals (green) overlapped. Moreover, the proportion of overlap accounting for positive AmpFISH cells (91.68 ± 2.07%) was slightly lower than that for positive IF signal (98.50 ± 1.14%) (Fig. 5c), suggesting that the sensitivity of AmpFISH should be better than that of IF in our workflow.

To demonstrate the ability of AmpFISH to identify the neuronal cell population, we used the established AmpFISH/IF workflow to detect the proportion of subpopulation in the two midbrain regions, the ventral tegmental area (VTA) and substantia nigra pars compacta (SNC), respectively. Tyrosine

hydroxylase (Th) protein and CaMKIIα mRNA were detected simultaneously in the same slice from the two midbrain regions, as shown in the lower panels of Fig. 5b. We compared the proportion of $Th^+/CaMKIIα^+$ subpopulation to $Th^+$ population in the two brain regions. The results showed that this proportion was significantly higher in the VTA area (25.27 ± 0.46%) than in the SNC area (13.97 ± 1.04%) (Fig. 5d). The projections and functions of neurons correlate with their expressed molecules in certain cell types[24,25]. Whether different proportions of $Th^+/CaMKIIα^+$ cells in VTA and SNC correlate with their functions in different brain regions needs to be further studied.

## Discussion

In this study, we provided a methodology to detect transcripts and chromosomes with a high imaging efficiency and simple workflows. We first developed a CRCA to prepare lssDNAc that could bind to a high number of fluorescent oligo-probes. Based on lssDNAc, we established simple workflows for AmpFISH that are easily implemented. We used Bst 2.0 DNA polymerase to determine CRCA. However, phi29 DNA polymerase was unable to form fragments of appropriate lengths due to its powerful extension ability. We tested various durations of DNA extension (0.5–2.0 h) and reaction temperatures (45–55 °C) using Bst 2.0 DNA polymerase. We found that the amplification probes with 500–1000 bp could effectively amplify the signal. The ssDNAc with over 1000 bp could probably produce a brighter signal than the ssDNAc with 500–1000 bp; however, these results need to be further investigated.

Previously published studies reported that the padlock DNA probe could be efficiently extended in situ by phi29 DNA polymerase via RCA, in which the enzyme was added to tissues[26]. However, the in situ signal amplification of RCA may be limited. To improve the SNR following in situ amplification, the brain slices usually require further tissue clearing and hydrogel

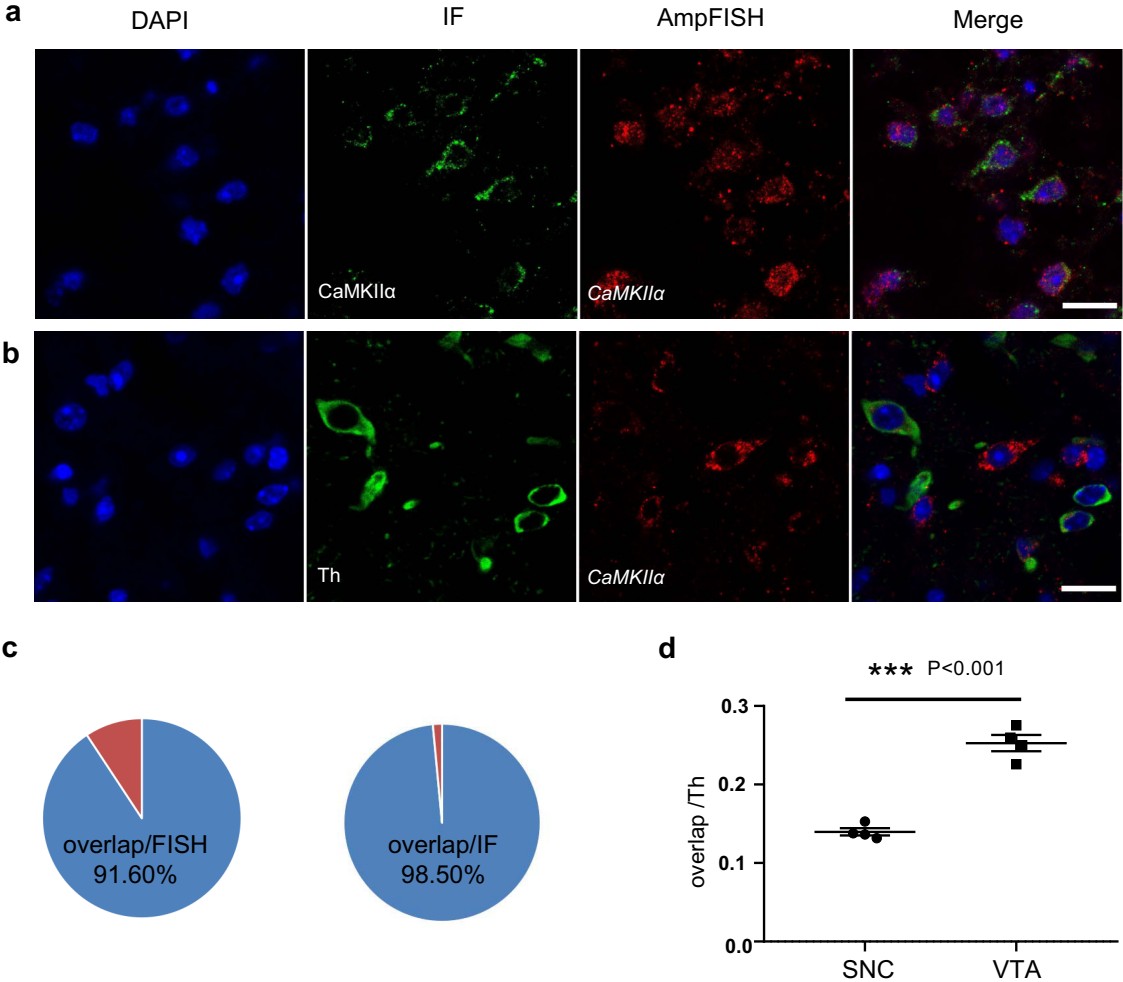

**Fig. 5 AmpFISH assay in combination with IF. a** IF and AmpFISH assays were verified mutually. IF for CaMKIIα protein and AmpFISH for CaMKIIα mRNA were double-labeled in the same slice from the CPu brain region (scale bar: 20 μm). **b** The TH + CaMKIIα⁺ cell subtypes were detected in the slice from the VTA brain region through a combination of IF with AmpFISH (scale bar: 20 μm). **c** The proportion of positive double-labeled cells was analyzed. "Overlap" represents cells with both IF-and AmpFISH-positive signals. Overlap/FISH represents the proportion of overlapping cells with a positive AmpFISH signal (91.68 ± 2.07%, $n = 5$). In the same way, Overlap/FISH represents the proportion of overlapping cells with a positive IF signal (98.50 ± 1.14%, $n = 5$). The total number of "overlap" cells, cells with AmpFISH signal, and cells with IF signal were 170, 185, and 172, respectively, from five different brain slices. **d** Statistical analysis and comparison of cell populations in the SNC and VTA brain slices. The proportion of Th⁺/CaMKIIα⁺ cells ("overlap") accounting for T⁺ cells in the VTA area (25.27 ± 0.46%) is significantly higher than that in the SNC area (13.97 ± 1.04%) ($n = 4$) (unpaired $t$-test with Welch's correction, two-tailed, $P < 0.001$).

protocols to improve imaging resolution. However, this process could perturb tissue components. AmpFISH overcame the problem of in situ amplification and efficiently enhanced the SNR through branching amplification, with no requirement for further tissue clearing and hydrogel protocols. A study had reported a FISH method that produced amplified signals via the HCR strategy, and allowed distinguishing single-nucleotide variations within mRNA molecules, while its application was demonstrated only at the cell level[27]. In comparison, AmpFISH in this study provided wider applications in both cells and tissues, although AmpFISH could not detect single-nucleotide variations.

Each 200 μL of the amplification system was used to extend the lssDNAc, and 1.2 μL (100 μM) of the adaptor and 1 μL (100 μM) of padlock oligonucleotide were added to the system, respectively. The average amount of recycling lssDNAc was 3–5 μg. A sample of the FISH experiment required 0.5 μg for each lssDNAc, meaning that one tube of ssDNAc probes can be used for 6–10 samples. The cost of ssDNAc preparation ranged from ¥2 to ¥15 ($0.28–$2.1) per assay (Supplementary Note 1), which is

dramatically lower than the cost of commercial probes. Moreover, we used 1 mL of the hybridization buffer to hybridize the samples. We will further develop a hybridization workflow with a smaller volume buffer for hybridization to increase its cost-effectiveness.

Penetrants, such as SDS, methanol, $H_2O_2$, HCl, and proteinase K, improve the permeability of the tissue. However, we found that an excellent effect cannot be achieved if all reagents were added. The excessive penetration may also lead to the loss of RNA. Another important factor for AmpFISH is PFA fixation. Both over fixation and no fixation should be avoided. Pre-fixation is required in our workflows during transcardial perfusion for tissue AmpFISH. Post-fixation could be avoided by conducting Amp-FISH in the normal tissue (Supplementary Fig. 7). However, when tissues marked with fluorescent protein are used, post-fixation is required, otherwise, the fluorescent protein might be lost. Although post-fixation for 1–2 h can greatly reduce the AmpFISH signal intensity, it is still acceptable (Supplementary Fig. 5). In addition, the sample should always be attached to the glass slide; otherwise, it cannot be used. For instance, in the

previously described IF/FISH protocol[14], in which proteinase K was directly used to digest the tissue slices after sectioning, we found it was difficult to attach the tissue slices to the glass slide after digestion by proteinase K. In our developed workflow, one more 10 min PFA fixation was performed after slice sectioning, easily allowing the binding of the tissue slice to the glass slide. To increase the stability of the assay, we used salmon sperm DNA as a competitor of the AmpFISH probes, but not synthetic oligonucleotides. However, synthetic oligonucleotides are theoretically better. When the background signal is high, the use of an appropriate concentration of oligonucleotides can be helpful.

The AmpFISH workflows are compatible with fluorescent proteins, which improve its application in certain scenarios. For example, in combination with transgenic mice expressing GFP, AmpFISH can determine the gene expression by overlapping the signal. In addition, the spatial distribution of molecules can be obtained by studying the GFP contour. Although the signal intensity of GFP is partly reduced after penetration, most signals of fluorescent protein can still be detected. Proteinase K should be avoided in the protocol, because the GFP signal will be lost completely. We penetrated the tissue by adding TritonX-100 and SDS instead of proteinase K. In contrast, moderate proteinase K was used to repair the antigen but not SDS in AmpFISH/IF experiment.

We noted that the sensitivity of IF greatly depends on its corresponding antibody, which may vary in titers when they are from different manufacturers. For example, we used two kinds of anti-Th antibodies from different manufacturers, which exhibited a significant difference in the IF signal for Th (Supplementary Fig. 8). In contrast, AmpFISH provided a more stable assay than IF and it does not depend on antibodies that are not always commercially available, making it a promising alternative under certain situations. Although some signal-enhanced methods for IF, such as Immuno-SABER[28], Immuno-RCA[29], etc. were developed, all of them require antibodies for specific proteins, which are not always commercially available or work stably. In contrast, AmpFISH probes are much easier and cheaper to be synthesized. Moreover, AmpFISH exhibited sensitivity and accuracy in both cells and tissues, and it is potentially suitable for imaging of multiplex transcripts. Further work is required to explore these possibilities. Altogether, AmpFISH is a convenient and versatile tool for FISH detection at a high level, offering a unique practical tool for developmental biology research, disease prognosis, and novel therapeutic developments.

## Methods

**Mice**. All animal procedures were conducted according to the Guide for the Care and Use of Laboratory Animals of Huazhong University of Science and Technology and approved by the Animal Care Committee of Hubei Province, China. Animals were housed individually in a 12 h (7 a.m.–7 p.m.) light/dark cycle, with food and water ad libitum. To avoid the circadian expression of gene, we sacrificed mice at approximately 10 a.m. each time. The mice used for Fig. 4d were transgenic mice of Tg(Thy1-EGFP)MJrs (Jackson Labs stock #007788) (8–9 weeks). In other experiments, mice used were male C57/BL6 (8–9 weeks).

**NIH3T3 cell and HeLa cell culture**. NIH3T3 and HeLa cells were purchased from Wuhan University Preservation Center (Wuhan, China). All cells were grown in Dulbecco's modified Eagle medium (DMEM, Gibco) supplemented with 10% fetal bovine serum (FBS, Sigma), 50 U/mL penicillin, and 50 µg/mL streptomycin (Gibco, cat. no.15070) at 37 °C under 5% $CO_2$. To transiently transfect EYFP in HeLa cells, we used polyethylenimine[30] (25 kd, Polysciences, Inc., cat. no. 23966) as the delivery reagent to transfect the pCAG-EYFP plasmid. After 3 days of transfection, the transfected HeLa cells were seeded onto glass slips. The transfected cells attached to the glass slips were used for conducting AmpFISH experiments.

To compare the signal magnification of AmpFISH (Fig. 2b), we established a stable NIH3T3/Cas9 cell line. The stable NIH3T3/Cas9 cell line was transfected with pX330-Cas9-2A-mCerulean-2A-puroR plasmid (note: mCerulean is the variant of cyan fluorescent protein and puroR is puromycin resistance gene; the maps are shown in Supplementary Fig. 3). After 1 month of puromycin screening, the stable NIH3T3/Cas9 cells with cyan fluorescent signal were established. We

seeded these cells on round glass slips with 18 mm diameter that can be placed inside 12-well plates. When the cell density reached 60–80% confluency, the cells were used for FISH.

**FISH probe design**. In order to design amplification probes, the toehold sequences in the adaptor were designed using the Picky 32-bit versions software[31]. The sequences of mouse transcripts (Mus_musculus.GRCm38.cdna.all.fa.gz) were downloaded from ftp://ftp.ensembl.org/pub/release-102/fasta/mus_musculus/cdna/. The parameters of Picky were set as GC content 30–70%, oligo size 30–40 for toehold part (Supplementary Note 2), and $T_m$ value >70 °C. The complementary sequence to the padlock circle was added to the 3′ end of the toehold sequence. Finally, the whole adaptor sequences were further assessed by the UNPACK online software[32] (http://www.nupack.org/partition/new). To minimize the probability of secondary structures, we tried to avoid using guanine in the padlock. The $T_m$ values were calculated using the following formula[33]:

$$T_m = 81.5 + 16.6\,(\log M) + 0.41\,(\%G + C) - 0.72\,(\%\,\text{formamide}),$$

where $T_m$ is the melting temperature in degrees Celsius, M is the monovalent salt molarity, (% G + C) is the percentage of guanine and cytosine in the DNA strand of interest, and (% formamide) is the percentage of formamide added.

**Preparation of FISH probes**. The oligonucleotide and fluorophore-tagged oligonucleotides (see Supplementary Table) were synthesized by Tianyi Huiyuan Biotech. Ltd. Wuhan, China. The synthesized ssDNA of adaptors were purified by desalting, the synthesized ssDNA of padlock with 5′ phosphorylation were purified by high-performance liquid chromatography (HPLC). The adaptor (0.6 µL, 100 µM) and padlock ssDNA (0.5 µL, 100 µM) were mixed and added to the EP tube with 42.9 µL of 1× annealing buffer (10 mM Tris, 50 mM NaCl, pH 8.0). The EP tubes with the mixture were put in a dry bath incubator (MK200–2, Allsheng Instruments Co., Ltd., Hangzhou, China) at 95 °C for 3–5 min. Next, the power to the incubator was turned off to spontaneously cool down the mixture until the temperature reached room temperature (RT). Then 5 µL of 10×T4 ligase buffer and 2 µL of T4 DNA ligase (Takara Corp., Dalian, China) were added to the mixture. The total volume was made to 50 µL by adding ddH₂O. After overnight ligation reaction, 5 µL of 10×Bst 2.0 polymerase buffer, 3 µL of Mg₂SO₄ (8 mM), 3 µL of dNTP (10 mM/each), 1 µL of Bst 2.0 DNA polymerase (New England Biolabs, Ipswich, MA, USA), and 38 µL of ddH₂O were added to form the extension reaction system with 100 µL reaction volume. The extension reaction was set for 2 h at 50 °C. After the extension, LssDNAcs were separated by electrophoresis in 1.0% agarose gel using the GelRed dye. Next, DNA gel images were acquired using the Bio-Rad Universal Hood II imaging system. The DNA bands were excised from the agarose gel with a clean, sharp scalpel. DNA was extracted using the QIAquick Gel Extraction Kit following the manufacturer's protocol. The purified ssDNAs were analyzed and quantified by NanoDrop™ 2000/2000c spectrophotometer.

**Coverslip preparation**. Glass coverslips were placed in 12-well plates and soaked in 0.1% diethylpyrocarbonate (DEPC) water overnight. Next, cells were washed with autoclaved DEPC water. Next, the coverslips were treated with methacryloxypropyltrimethoxysilane (Bind-Silane) for 1 h and further treated with the poly-L-lysine solution for 5 min. The poly-L-lysine solution was removed and coverslips were dried at 60 °C for 1–2 h.

**FISH in fixed cell chambers**. To fix the cultured cells, cells were rinsed in 1× phosphate-buffered saline (PBS) at RT, and then immediately fixed in 4% PFA for 10 min. Finally, the cells were rinsed in 1× PBS. For permeabilization, the cells were washed with the PBST buffer (1× PBS with 0.5% TritonX-100) for 10 min, then washed with 1× PBSTw (1× PBS with 0.1% Tween-20) for 1 min, and subsequently washed with 2× SSC/TritonX-100 (2× SSC with 0.1% Triton X-100) for 2 min. The primary amplification probe sets were denatured at 95 °C for 3–5 min. Each primary amplification probe at 0.5 µg/mL final concentration was added to the hybridization solution (2× SSC, 1% Triton X-100, 40% formamide, 10% dextran sulfate, 20 mM RVC, 0.1 mg/mL salmon sperm DNA). The cells were gently shaken overnight at 42 °C and then washed with 1× PBS containing 40% formamide (3 × 10 min) and then with 2× SSCT buffer (3 × 10 min) at 42 °C. For secondary or tertiary amplification hybridization, the probes were added to the hybridization solution (2× SSC, 1% Triton X-100, 20% formamide, 10% dextran sulfate), 20 mM RVC, 0.1 mg/mL salmon sperm DNA, and 0.5 µg/mL of each amplification probe with gentle shaking at 37 °C for 4–5 h. Next, the cells were washed with 1× PBS containing 20% formamide (3 × 10) min and then with 2× SSCT (3 × 5 min) at 37 °C.

For fluorescence detection, tissue slices were rinsed once in 2× SSCT at RT and then stained with DAPI (Invitrogen, no. D1306) for 30 min. Next, the samples were washed with 1× PBS (3 × 10 min) containing 0.1% TritonX-100 at RT. Fluor hybridization solution (2× SSC, 20% formamide, 10% dextran sulfate, 150 nM fluorescent probes) was added and held at 37 °C for 30 min. Then, samples were washed with 2× SSC (3 × 10 min) containing 20% formamide at 37 °C. The round glass slides with samples were placed into a magnetic chamber (L-shape tubing type Chamlide CM-B18-1, Live Cell Instrument, Seoul, South Korea) for imaging.

**UMI-mRNA sequencing**. NIH3T3 cells were grown in DMEM (Dulbecco's Modified Eagle Medium, Gibco) supplemented with 10% FBS (Fetal Bovine Serum, Sigma), 50 U/mL penicillin, and 50 µg/mL streptomycin (Gibco,cat.no.15070) at 37 °C with 5% $CO_2$. Cells were treated with 0.25% trypsin solution (HyClone, No.SH42605.01) when they reached ~$10^6$ cells/mL. Then, the cells were washed with 1X PBS, and then mixed with 1 mL TRIzol solution (ThermoFish, No.15596029), and snap-frozen with dry ice. Total RNA was qualitatively and quantitatively evaluated as follows: (1) the RNA sample was initially qualitatively evaluated using 1% agarose gel electrophoresis for possible contamination and degradation; (2) RNA purity and concentration were then examined using NanoPhotometer® spectrophotometer; (3) RNA integrity and quantity were finally measured using RNA Nano 6000 Assay Kit for the Bioanalyzer 2100 system. After library preparation and pooling of different samples, the samples were subjected to Illumina sequencing. The libraries were sequenced using the Illumina NovaSeq 6000 Platform for 6 G raw data and generated 150 nt paired-end reads. UMI sequences on each read were identified by UMI-tools (1.0.0), and reads with UMIs were used for the subsequent analysis. To identify the duplicated reads, UMIs were initially removed from the UMI reads, and the remaining parts of each read were mapped to the reference genome using Hisat2. Reads that mapped to the same location on the reference genome were identified as duplicated reads. Then, the UMIs on each read were recalled, and the duplicated reads with the same UMI were identified as non-natural duplications, which were subsequently removed from the processed data. HTSeq v0.6.1 was used to count the read numbers mapped to each gene. Then, the FPKM of each gene was calculated based on the length of the gene, and the read count was mapped to the gene. The sequence data were uploaded onto GEO database (GEO accession numbers: GSE181685).

**FISH for centromere detection in HeLa cells**. HeLa cells were seeded in 12-well chambers. After growing to 50–70% confluency, HeLa cells were rinsed with 1× PBS and then fixed by 4% PFA solution for 10 min. Next, the cells were rinsed with 1× PBS. The cells were treated with 2× SSCT containing 70% formamide at 70 °C (3 min) and then transferred to ice-cold 70% ethanol (5 min), 90% ethanol (5 min), and 100% ethanol (5 min) successively.

After fixation, the cells were rinsed with 1× PBS (1 min), next with 1× PBST (1× PBS/0.5% Triton X-100) for 10 min, then with 1× PBSTw (1× PBS buffer/0.1% Tween-20) for 2 min. The cells were incubated in 0.1 N HCl (5 min) and washed with 2× SSC/0.1% TritonX-100 (2 × 5 min). Next, the cells were incubated in 2× SSCT with 50% formamide (10 min) and transferred to fresh 2× SSCT with 50% formamide at 60 °C (at least 1 h). The wells were loaded with 500 µL of a solution containing 2× SSCT, 50% formamide, 10% (wt./vol.) dextran sulfate, and 400 ng/µL RNase A, primary probe (final concentration 500 ng/mL, denaturation at 80 °C for 3 min). The samples were incubated overnight at 44 °C. After hybridization, samples were washed with 2 × SSCT at 60 °C (3 × 10 min). The samples were stained with DAPI and washed with 1× PBS (3 × 10 min).

The samples were incubated with fluorescent probes (150 nM final concentration) in the fluorescent hybridization buffer (1× SSC, 35% formamide, 10% dextran sulfate at 37 °C (30 min). The cells were then washed (3 × 10 min) at 37 °C with wash buffer (1× SSC, 35% formamide).

**FISH in fixed tissue chambers**. The round glass slips with 18 mm diameter were autoclaved at 210 °C for 8 h. The glass slips were treated with methacrylox-ypropyltrimethoxysilane (Bind-Silane) and were further treated with poly-L-lysine solution. Adult mice were anesthetized with a mixture solution of 2% chloral hydrate (160–200 mg/kg) and xylazine (15–20 mg/kg), and transcardially perfused with 1× DEPC-PBS for 10 min, followed by treatment with 4% PFA in PBS. The brains were removed and immersed in 30% sucrose overnight at 4 °C until the brains sunk to the bottom. Then brains were placed onto OCT and frozen at −80 °C with isopentane. Next, 16–20 µm sections were prepared using a cryostat (Leica CM1700). The brain slices were attached to the pretreated glass slips, then fixed with 4% PFA in PBS at RT for 10 min, dehydrated through gradient methanol/PBS (50%–75%–95%) for 3 min each, treated with methanol at −20 °C overnight, and then placed at −80 °C for 15 min before hybridization.

For hybridization, slices were taken from −80 °C and equilibrated to RT for 5 min. These slices were washed with 1× PBS/0.1% TritonX-100 (3 × 5 min) containing 0.1 U/µL recombinant RNase inhibitor (RRI, Takara corp. in Dalian, China). Primary amplification probes were denatured at 60 °C for 3–5 min and then directly added to the hybridization buffer (2× SSC, 1% Triton X-100, 40% formamide, 10% dextran sulfate, 20 mM RVC, 0.1 mg/mL salmon sperm DNA, and 1% SDS). The final concentration of each primary amplification probe was 0.5 µg/mL. The slices were incubated at 42 °C with gentle shaking overnight, and then washed with 1× PBS/40% formamide (3 × 10 min) at 42 °C and then with 2× SSC/0.1% Triton X-100 at RT. For secondary amplification hybridization, secondary amplification probes were denatured at 90 °C for 3–5 min and were added to the hybridization solution (2× SSC, 1% Triton X-100, 25% formamide, 10% dextran sulfate, 20 mM RVC, 0.1 mg/mL salmon sperm DNA, 1% SDS) to form 0.5 µg/mL for each probe. Samples were incubated for 4–5 h with gentle shaking at 37 °C. Samples were washed 3 × 10 min with 1× PBS containing 25% formamide at 37 °C. The samples were then washed with 2× SSCT (3 × 5 min) at 37 °C and 2× SSC/0.1% TritonX-100 (3 × 5 min) at RT.

For fluorescent hybridization, samples were rinsed in 2× SSCT and then stained with DAPI (Invitrogen, no. D1306) for 1 h at RT. The samples were washed with 1× PBS (3 × 10 min) with 0.1% TritonX-100 at RT. Fluor hybridization solution (2× SSC, 20% formamide, 10% dextran sulfate), and 150 nM fluorescent probes were incubated at 37 °C for 30 min. The samples were washed 3 × 10 min with wash buffer (2× SSC/20% formamide) at RT. Round glass slides with samples were mounted onto a magnetic chamber (L-shape tubing type Chamlide CM-B18-1, Live Cell Instrument, Seoul, South Korea) for imaging.

**IF and FISH**. The animal brain slices were obtained and dehydrated as described above. The sections were taken from −80 °C, equilibrated to RT for 15–20 min, and washed with PBSTR (1× PBS/0.1% TritonX-100, 0.1 U/µL RNase inhibitor) for 3 × 5 min. The sections were then treated with protein K (1.5 ng/mL) at RT for 15 min, HCl (0.1 N) at RT for 20 min, and 2% $H_2O_2$ at RT for 20 min, washed with PBSTR for 5 min, followed by incubation in 1× first hybridization buffer (40% formamide, 2×SSC, 1% TritonX-100, 20 mM RVC, 0.1 mg/mL salmon sperm DNA, 10% dextran sulfate, and probes at 500 ng/mL per oligo) at 42 °C with gentle shaking overnight. Then, the sections were washed with 40% formamide buffer (40% formamide/2× SSC) at 42 °C for 20 min thrice and PBSTR for 5 min, followed by incubation in 1× second hybridization buffer (25% formamide/2× SSC, 1% TritonX-100, 20 mM RVC, 0.1 mg/mL salmon sperm DNA, 10% dextran sulfate, and probes at 0.5 µg/mL per oligo) at 37 °C with gentle shaking overnight. The next day, the sections were washed with 25% formamide buffer (2× SSC, 25% for-mamide) at 37 °C for 3 × 20 min and PBSTR for 5 min, and then incubated with DAPI at RT for 30 min. Next, the sections were washed with PBS at RT (3 × 10 min). The sections were incubated in the fluorescent hybridization solution (2× SSC, 20% formamide, 10% dextran sulfate, 150 nM fluorescent imager strands) at 37 °C for 30 min, and washed with 20% formamide buffer (20% formamide, 2× SSC) at 37 °C for 3 × 10 min.

For IF labeling, the sections were incubated with blocking buffer (1× PBS with 0.3% TritonX-100 and 10% goat serum) at RT for 30 min. Primary antibody for CaMKIIα was purchased from GeneTex Inc. (No. GTX127939), and anti-Th antibody was purchased from Sigma-Aldrich (No. AB152) and Abcam (No. ab6211). Primary antibodies were diluted at a ratio of 1:500 with blocking buffer, and samples were incubated overnight at 4 °C. Next, samples were washed with 1× PBS (3 × 10 min) and then labeled with secondary antibody (goat anti-rabbit Alexa 488, 1:500) for 2 h at RT. Finally, the samples were washed with 1× PBS (3 × 10 min).

**Imaging set-up**. For FISH imaging in HeLa and stable NIH3T3/Cas9 cells, the data were acquired by Nikon Ni-E microscopy with a 25× water-immersed objective lens (numerical aperture: 1.1); 405 and 561 nm lasers were used. A set of filters UV-2E/C (EX: 325–375, DM: 400, EM: 425–475) was used to visualize DAPI staining. G-2E/C (EX: 528–553, DM: 565, EM: 570–620) was used to visualize the TAMRA dye signal.

For GFP FISH imaging in HeLa cells and FISH/IF imaging in brain slices, data were acquired by Zeiss LSM 710 microscope; 405, 488, and 561 nm lasers were used. The following set of filters were used: filter set 49 (EX: G365, DM: 395, EM: 422–498) for DAPI dye, filter set 38 (EX: 470/40, DM: 495, EM: 495–553) for GFP fluorescent protein and Alexa 488, and filter set 43 (EX: 545/25, DM: 570, EM: 563–660) for TAMRA dye signal. Color images were captured with a Zeiss AxioCamICc5 camera.

**Statistics and reproducibility**. The signal puncta were quantified using the Fiji software (NIH). The signal intensity of puncta was quantified in a 16-bit figure using a previously described method[34]. Labeled neurons from each sample were randomly selected. Using the selection tool of ImageJ software, an area covering the signal spot was selected within the defined region, and the integrated density (IntDen) of the signal was measured using an ImageJ plugin. To measure the background signal, a straight line was brought across the background area close to the signal spot, and the mean gray value was measured. The total mean value of the background of the rectangular selection was subtracted from the integrated density of the signal to obtain the total signal density in the rectangular selection. The average intensity of the signal without background was calculated by dividing the total signal density by the pixel number of the rectangular selection.

$$\text{Signal intensity} = [\text{IntDen} - (\text{area} \times \text{mean of background})]/\text{Area}$$

For statistical analysis, one-way analysis of variance (ANOVA) followed by Bonferroni correction was performed using GraphPad Prism 5 (Fig. 4c) and an unpaired t-test with Welch's correction (Fig. 5c). Differences between the datasets were considered significant at $P < 0.05$.

**Reporting summary**. Further information on research design is available in the Nature Research Reporting Summary linked to this article.

## Data availability

Sequencing data that support the findings in this study have been assigned Gene Expression Omnibus accession number GSE181685. Source data for the graphs and charts can be found in http://figshare.com/s/51ea12e69cbdfca79e09. All other data are available from the corresponding author on reasonable request.

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

## Acknowledgements

We thank the Optical Bio-imaging Core Facility of WNLO-HUST for the support in data acquisition. This work was supported by the Major Program of National Natural Science Foundation of China [Grant No. 81827901], National Natural Science Foundation of China (Grant No. 91749209).

## Author contributions

D.C., S.W. and C.X. carried out mRNA hybridization in brain slices and cells, and FISH imaging; S.W. prepared the FISH probes, D.L. conducted the DNA hybridization, K.Z. screened the stable cell line, J.Y. designed the experiments and carried out data analysis and wrote the manuscript. Z.Z., H.G. and Q.L. participated in planning of experiments, discussion, and contributed to the preparation of the manuscript. Q.L. and J.Y. supervised the project.

## Competing interests

J.Y., D.C., S.W. and C.X. have filed a provisional patent on this technology. The authors declare no competing interests.
