## [Peer Review File. · Communications Biology]

Reviewers' comments:

Reviewer #1 (Remarks to the Author):

In this study, Cao et al., present a method to prepare long single-strand DNA concatemers (lssDNAC) through a controllable rolling-circle amplification (CRCA). The manuscript is clear, organized, showing the advantages of AmpFISH in the detection of specific targets.

Major comments:

Although the authors analyzed the specificity of the method by positive/negative cells (e.g Figure 2A), a detailed analysis of the sensitivity/specificity of the methods should be performed. This could be performed in completely negative cells lines/tissue samples and having transcripts that vary in some nucleotides.

Authors should include the reference from Salvatore A et al., PNAS, 2019 and discuss differences between both methods.

Minor comments:

Supplementary Fig. 3 showing the comparison between AmpFISH imaging with primary amplification probes versus secondary amplification probes via branching amplification is not clear. In figureA, although there is some background there is a positive signal as well. The authors should explain the scientific reason behind the increased background with primary amplification. Supplementary video is missing, please add to the files.

The authors should discuss why to include full DNA and none synthetic nucleic acids to increase the stability of the assay.

Reviewer #2 (Remarks to the Author):

Cao, Wu et al. have developed a modified version of AmpFISH technology which employs rolling circle amplification as opposed to the self-assembling hairpin probes. The authors demonstrated that the method appears to be suitable for labeling to both individual mRNA molecules as well as chromosomes. The authors have also optimized the method to ensure that it can be multiplexed with immunofluorescence. This study represents an incremental development in FISH technology that could find a broad and interested audience. I have attached some comments and suggestions that I believe would help to further expand the number of potential users of this method.

1. As this is effectively a methods paper, it would be very useful to highlight some more potential points of optimization. For example, parameters such as fixative, length of fixation, permeabilization strategy, and hybridization strategy can vary between cell lines and tissue types. This optimization was provided for clearing of brain slices which was useful. If the authors have used the protocol in multiple cell lines, tissue types, or organisms, then it would be very useful to advise readers how parameters may change between sample types. This would improve reproducibility and encourage many potential users to apply the method in samples other than HeLa cells or brain slices.

2. It was helpful to include the data to justify the decision on which polymerase and temperature to use. In line with comment #1, it may be useful to discuss other points of optimization. As interested users may have to modify the protocol to different types of tissue preparation, it is good to have access to this information.

3. Please clarify using ~1 sentence how the signal-to-noise ratio was determined in Figure 2C and 4C. This is described in more detail in the methods section, but it would be useful to see directly in the caption that the intensity of an area next to the signal was subtracted from the signal itself. The y-axis in these figures could be changed from "SNR" simply to "signal-to-noise ratio" to improve readability of the figure panels.

4. Line 148: could specify that primary amplification was used for this example.
5. Line 199: specify the concentration of proteinase K used, this has a massive impact on morphology.
6. Line 256: should read "and up to 1 h post-fixation is acceptable".
7. Line 326: abbreviation should be HPLC not HLPC.
8. Line 342: "Coverslip preparation" might make more sense.
9. The inclusion of the sequences for target mRNA and oligo design in the supplementary info is very useful and will encourage users to adapt the protocol.

Reviewer #1 (Remarks to the Author):

In this study, Cao et al., present a method to prepare long single-strand DNA concatemers (lssDNAc) through a controllable rolling-circle amplification (CRCA). The manuscript is clear, organized, showing the advantages of AmpFISH in the detection of specific targets.

Major comments:

Although the authors analyzed the specificity of the method by positive/negative cells (e.g Figure 2A), a detailed analysis of the sensitivity/specificity of the methods should be performed. This could be performed in completely negative cells lines/tissue samples and having transcripts that vary in some nucleotides.

Authors should include the reference from Salvatore A et al., PNAS, 2019 and discuss differences between both methods.

Response: To analyze the sensitivity and specificity of the AmpFISH method, we sequenced the NIH3T3 cell line via UMI-RNAseq experiments. Based on the sequencing results, we selected four genes with different expression levels. Then, we designed the AmpFISH probes for targeting those mRNAs. The AmpFISH results were consistent with the UMI-RNAseq results that were shown in Figure 2 b and Figure 2c of the revised manuscript. Moreover, the results of Figure 5a and 5c show that up to 90% of AmpFISH signal of *CaMKII α* can overlap with IF signals of *CaMKII α* in the brain tissues. Those results demonstrated high sensitivity and specificity of AmpFISH.

Based on the study by Salvatore A et al., PNAS, 2019, we discussed the differences between both methods in the line 202-206 of the revised manuscript: A study had reported a FISH method that produced amplified signals via the HCR strategy, and allowed distinguishing single-nucleotide variations within mRNA molecules, while its application was demonstrated only at the cell level³⁴. In comparison, AmpFISH in this study provided wider applications in both cells and tissues, although AmpFISH could not detect single-nucleotide variations.

Minor comments:

1. Supplementary Fig. 3 showing the comparison between AmpFISH imaging with primary amplification probes versus secondary amplification probes via branching amplification is not clear. In figureA, although there is some background there is a positive signal as well. The authors should explain the scientific reason behind the increased background with primary amplification.

Response: In Supplementary Figure 3 of the previous version of the manuscript, some backgrounds showed bright signals. We used the brain slice from the striatal area, which is rich in nerve fibers. We speculated that the background signal with primary amplification was due to the binding of the hydrophobic fluorescent probe to the nerve fiber. When the positive signal was weak, the background became more conspicuous when we adjusted the figure contrast.

To avoid misunderstanding among the readers, we repeated the experiments using mice cortex with lesser nerve fibers. We found that the bright background disappeared. We have now replaced the figure in Supplementary Figure 3.

2. Supplementary video is missing, please add to the files.

Response: Thank you for pointing out the error. The supplementary video has been provided with the revised version of the manuscript.

3. The authors should discuss why to include full DNA and none synthetic nucleic acids to increase the stability of the assay.

Response: Thank you for the excellent suggestion.

To discuss this, we have added the following sentences in the revised manuscript: "To increase the stability of the assay, we used salmon sperm DNA as the competitor of the AmpFISH probes, but not synthetic oligonucleotides. However, synthetic oligonucleotides are theoretically better. During high background signals, the use of an appropriate concentration of oligonucleotides could be helpful."

Reviewer #2 (Remarks to the Author):

Cao, Wu et al. have developed a modified version of AmpFISH technology which employs rolling circle amplification as opposed to the self-assembling hairpin probes. The authors demonstrated that the method appears to be suitable for labeling to both individual mRNA molecules as well as chromosomes. The authors have also optimized the method to ensure that it can be multiplexed with immunofluorescence. This study represents an incremental development in FISH technology that could find a broad and interested audience. I have attached some comments and suggestions that I believe would help to further expand the number of potential users of this method.

1. As this is effectively a methods paper, it would be very useful to highlight some more potential points of optimization. For example, parameters such as fixative, length of fixation, permeabilization strategy, and hybridization strategy can vary between cell lines and tissue types. This optimization was provided for clearing of brain slices which was useful. If the authors have used the protocol in multiple cell lines, tissue types, or organisms, then it would be very useful to advise readers how parameters may change between sample types. This would improve reproducibility and encourage many potential users to apply the method in samples other than HeLa cells or brain slices.

Response: Thank you for the excellent suggestion. To make the protocol concise, we have provided the different steps of AmpFISH in Supplementary Figure 5 and Supplementary Figure 6. For optimization, we adjusted the PFA post-fixation time (Supplementary Figure 7) and the hybridization time of primary amplification probes (Supplementary Figure 8) in the mouse brain slices. We found that a high signal-to-noise ratio in the mouse brain slice was produced without post-fixation and primary amplification hybridization for 24 hours. We think that those two steps were crucial for optimizing other tissue or cell samples.

2. It was helpful to include the data to justify the decision on which polymerase and temperature to use. In line with comment #1, it may be useful to discuss other points of optimization. As interested users may have to modify the protocol to different types of tissue preparation, it is good to have access to this information.

Response: We have added the part in the Discussion section in the lines 189-195 of the manuscript: "We used Bst 2.0 DNA polymerase to determine CRCA. However, phi29 DNA polymerase was unable to form fragments of appropriate lengths due to its powerful extension ability. We tested various durations of DNA extension (0.5-2.0

hours) and reaction temperatures (45°C-55°C) using Bst 2.0 DNA polymerase. We found that the amplification probes with 500 to 1,000 bp could effectively amplify the signal. The ssDNAc with over 1,000 bp could probably produce a brighter signal than the ssDNAc with 500 to 1,000 bp; however, these results need to be further investigated.”

3. Please clarify using ~1 sentence how the signal-to-noise ratio was determined in Figure 2C and 4C. This is described in more detail in the methods section, but it would be useful to see directly in the caption that the intensity of an area next to the signal was subtracted from the signal itself. The y-axis in these figures could be changed from “SNR” simply to “signal-to-noise ratio” to improve readability of the figure panels.

Response: Thank you for the suggestion.

“The intensity of the background signal, from an area adjacent to that from which the signal was recorded, was subtracted from the intensity of the signal. The signal-to-noise ratio was determined using the formula: (Signal-background)/background.” We have added the sentence mentioned above in Figure 2c and Figure 4c.

4. Line 148: could specify that primary amplification was used for this example.

Response: Thank you for the suggestion. We have added the part in line 569 of the revised manuscript.

5. Line 199: specify the concentration of proteinase K used, this has a massive impact on morphology.

Response: Thank you for the suggestion. We have added the concentrations of proteinase K, HCl, and H₂O₂ in lines 167 of the revised manuscript.

6. Line 256: should read “and up to 1 h post-fixation is acceptable”.

Response: Thank you for pointing out the error. We have corrected the misspelled word in line 219-222 of the revised manuscript. To further increase the accuracy of the sentence, we have revised it as, “Post-fixation could be avoided by conducting AmpFISH in the normal tissue. However, when the tissues marked with fluorescent protein are performed, post-fixation is required, otherwise, the fluorescent protein might be lost. Although post-fixation for 1-2 h can greatly reduce the AmpFISH signal intensity, it is still acceptable (Supplementary Figure 7).”

7. Line 326: abbreviation should be HPLC not HLPC.

Response: Thank you for pointing out the error. We have corrected the abbreviation in line 294 of the revised manuscript.

8. Line 342: "Coverslip preparation" might make more sense.

Response: Thank you for the suggestion. We agree with the reviewer and have revised this part in line 309 of the revised manuscript.

9. The inclusion of the sequences for target mRNA and oligo design in the supplementary info is very useful and will encourage users to adapt the protocol.

Response: Thank you for the suggestion. We have provided the sequences of the target transcripts and oligo sequences in supplementary text 2 and the supplementary table of the revised manuscript.

REVIEWERS' COMMENTS:

Reviewer #1 (Remarks to the Author):

In this study, Cao et al., present a method to prepare long single-strand DNA concatemers (lssDNAc) through a controllable rolling-circle amplification (CRCA). Authors answer to all the reviews comments and add the missing information/experiments/references. The manuscript is now ready for publication.

Reviewer #2 (Remarks to the Author):

Cao, Wu, et al. have addressed my comments sufficiently. I have no additional concerns about this manuscript. However, I have included some small comments for the authors to consider.

1. Throughout the manuscript, the authors refer to 'high-level' FISH assays (e.g. lines 57 and 61). Could the authors elaborate what is specifically meant by high-level FISH? Otherwise I think that could be left out.

2. Line 74: typo, amply should be 'amplify'.

Reviewer #1

In this study, Cao et al., present a method to prepare long single-strand DNA concatemers (lssDNAc) through a controllable rolling-circle amplification (CRCA). Authors answer to all the reviews comments and add the missing information/experiments/references. The manuscript is now ready for publication.

Response: There is no question need to been answered. Thank you for reviewing and valuable advice.

Reviewer #2 :

Cao, Wu, et al. have addressed my comments sufficiently. I have no additional concerns about this manuscript. However, I have included some small comments for the authors to consider.

1. Throughout the manuscript, the authors refer to 'high-level' FISH assays (e.g. lines 57 and 61). Could the authors elaborate what is specifically meant by high-level FISH? Otherwise I think that could be left out.

Response: We have deleted "high-level" from the lines 57 and 61.

2. Line 74: typo, amply should be 'amplify' .

Response: We have corrected the wrong spelling. Thank you for reviewing and valuable advice.